

# Forest Fire Finder - DOAS application to long range forest fire detection.

Rui Valente de Almeida[1,2] and Pedro Vieira[1,2]

[1]NGNS - Ingenious Solutions, Rua Cidade de Évora, 11, 2660-022 Loures, Portugal
[2]Faculty of Science and Technology, NOVA University of Lisbon, Caparica Campus, 2829-516 Caparica, Portugal

*Correspondence to:* Pedro Vieira (pedro.vieira@ngns-is.com)

**Abstract.** Fires are an important factor in shaping Earth's ecosystems. Plant and animal life, in almost every land habitat, are at least partially dependant on the effects of fire. However, their destructive force, which many times has proven itself uncontrollable, is one of our greatest concerns, effectively originating several policies in the most important industrialised regions of the globe.

This paper aims to comprehensively characterise the Forest Fire Finder (FFF), a forest fire detection system based mainly upon a spectroscopic technique called Differential Optical Absorption Spectroscopy (DOAS). The system is designed and configured with the goal of detecting higher-than-the-horizon smoke columns by measuring and comparing scattered sunlight spectra. The article covers hardware and software, as well as their interactions and specific algorithms for day mode operation. An analysis of data retrieved from several installations deployed in the course of the last five years, is also presented.

Finally, this document features a discussion on the most prominent future improvements planned for the system, as well as its ramifications and adaptations, such as a thermal imaging system for short range fire seeking or environmental quality control.

## 1 Introduction

Fire is a process by which elements chemically combine with oxygen, releasing energy (as heat and light) and smoke into
the surrounding environment. Fires are an important factor in shaping Earth's ecosystems. Plant and animal life, in many land habitats, are at least partially dependant on the effects of fire (Food and Agriculture Organisation (FAO), 2007).

The use of fire by hominids pre-dates civilisation by thousands of years and, in today's society, there are almost no areas of technology or scientific knowledge that do not involve fire in one way or another. However, fire's destructive power is undeniable.

Forest fires are among the great concerns of the present day in industrialised countries. Research regarding wildfires has been targeted by many countries and unions worldwide, in an effort to minimise the negative impact these events imply.

In the European Union, the Horizon2020 research programme states that there must be a Union-wide investment in research concerning forest protection and recovery from fires. In the past, the FP7 programme had sponsored the development of an automatic forest fire detection system called FireSense, an investment of over 2.5 million euros (European Comission, 2012).



The United States Forest Service acknowledge the importance of understanding wildland fire dynamics, running a network of research centres solely dedicated to the study of this subject. Research endeavours take 6% of the Service's annual budget, which is currently directed primarily towards fire suppression (United States Forest Department, 2015).

Australia is another geographic region where wildfires have had a great impact. As a response, its government has created the Bushfire and Natural Hazards Cooperative Research Centre. The institution builds upon more than 10 years of experience dealing with Australian bushfires and aims to produce internationally recognised research regarding the study and modelling of wildfires in Australia and New Zealand (BNHCRC, 2016).

In spite of this global investigation effort regarding fires and their behaviour, every year, material losses as a result of fires ascend to billions of dollars and thousands of lives are lost in the same way. This leads to a strong increase in the size of the fire protection market, including passive and active detection platforms, which is expected to grow at a Cumulative Aggregate Growth Rate of 11,53% from 2014 to 2020 (Research and Markets, 2016).

## 2   State of the Art

In recent years, several methods have been developed in an attempt to automatically and reliably detect forest fires. These systems differ primarily in their strategical approach to the issue at hand, creating three main categories:

**Satellite Monitoring Techniques:**  Satellite data have been used for fire monitoring purposes since the late $20^{th}$ century. The MODIS (MODerate resolution Imaging Spectroradiometer) and AVHRR (Advanced Very High Resolution Radiometer) sensors, deployed respectively in the Aqua/Terra and NOAA satellites, have had extensive use in this regard. However, their low temporal resolution (2 and 4 times per 24 hours, respectively) make them poor candidates for fire detection uses. Geostationary satellites overcome this difficulty by continuously scanning a single, very large geographic region. They have, nevertheless, a low spatial resolution, of 1 km, which means that small fires are difficult for them to detect (Manyangadze, 2009).

**Wireless Network Sensing:**  The Wireless Sensor Network approach to fire detection is completely different from the other two categories. Instead of having a single device patrolling the target region, these systems are designed on the capabilities of a high number of extremely small battery operated sensor boards that can communicate among themselves (Alkhatib, 2014; Liyang Yu et al., 2005), as illustrated in Figure 1.

The sensor boards are equipped with several sensors, from temperature and humidity to luminance detectors. In spite of their great fire detection capabilities, these networks present various drawbacks, such as their very limited individual range of detection and their two year lifetime or the fact that their abandon might imply an environmental issue (Alkhatib, 2014).

**Large Area Remote Sensing:**  This family of systems is designed with the goal of minimising the number of deployed devices in a given target region. Their architecture implies the use of an optical principle in order to detect smoke or flames, whether optical cameras or spectrometers.



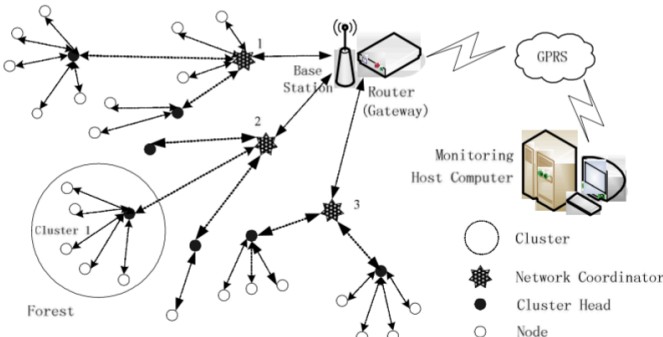

**Figure 1.** Example of a Wireless Sensor Networks general diagram (Alkhatib, 2014).

There are already several commercially available systems, such as the Forest Fire Finder (which is the main subject of this paper), FireWatch, ForestWatch, AlarmEYE or Eyefi SPARC. Although these are commercial products, the available information is sparse and many times outdated, so a true comparison not only is beyond the scope of the article but would also require more research efforts. Nevertheless, it is important to briefly describe the operating principles of the more prominent systems.

**ForestWatch:** developed in South Africa by EnviroVision Solutions, this system uses optical object recognition software, coupled to a very specific camera system. It detects smoke during the day and the flame glow during the night, at a maximum distance of 15 miles in every direction, in a semi-automatic fashion. It is probably the most commercially successful system, having more than 300 currently operating towers (EnviroVision Solutions; Hough, 2007).

**FireWatch:** FireWatch is a commercial system operated and sold by IQ Wireless Gmbh, in Germany. The system uses optical sensors and object recognition algorithms to detect smoke at a maximum distance of 15 km. It is important to mention that the FireWatch system is not a fully automatic fire detection platform, requiring a control room to operate correctly (IQ-Wireless, 2016).

**Forest Fire Finder:** the Forest Fire Finder (FFF) was developed in Lisbon, in a partnership between the NOVA University of Lisbon and NGNS-IS, Ltd., in 2006. This patented system uses a spectroscopic technique to assess the atmosphere and detect smoke columns (NGNS-IS, 2016). During the night, the system changes its operation mode and relies solely on image processing to detect a fire's glow. Its maximum rate detection range is of 15km, and it acts with complete autonomy, requiring minimal human intervention (see Figure 2).

The FFF system is the only one to use an optical spectroscopy technique to detect fire through smoke presence. Since the analysis is carried out in an outdoor scenario, the process is not as immediate and straightforward as in laboratory experiments. This article addresses only the spectroscopic techniques used in the system's daytime operation mode.





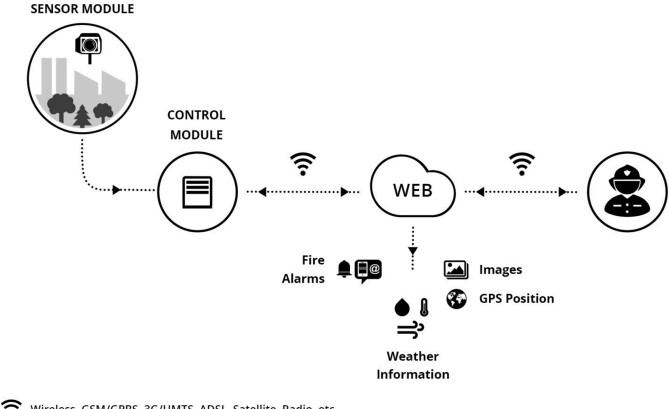

**Figure 2.** The Forest Fire Finder system (NGNS-IS, 2016).

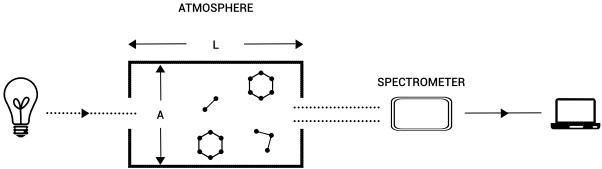

**Figure 3.** Active DOAS Schematic.

## 3   The Technique

The FFF system makes use of a spectroscopic technique called Differential Optical Absorption Spectroscopy (DOAS). This is a well established and widely used technique in the field of atmospheric studies (Platt and Stutz, 2007).

There are two main categories of DOAS experiment assemblies, with different goals and capabilities:

5    **Active Systems:**  These systems, of which a simple illustration is presented in Figure 3, are characterised by relying on an artificial light source for their measurements. A spectrometer at the end of the light path performs spectroscopic detection. Active DOAS techniques are very similar to traditional in-lab absorption spectroscopy techniques (Platt and Stutz, 2007);

**Passive Systems:**  Passive DOAS techniques, illustrated in Figure 4, use natural light sources, such as the sun and the moon, in

10    their measurement process. An optical system is pointed in certain elevation and azimuth angles and sends the captured light into a spectrometer, connected to a computer. The system returns the total value of the light absorption in its path (Platt and Stutz, 2007; Merlaud, 2013). Since the FFF system is basically a passive DOAS system, we will centre our discussion on this category from this point forward.





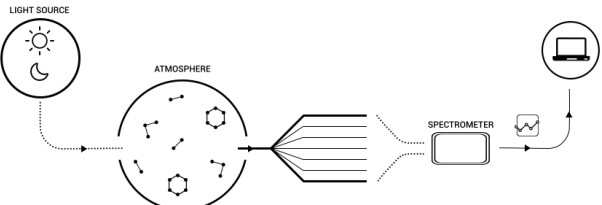

**Figure 4.** Passive DOAS Schematic

DOAS itself is based on Lambert-Beer's law, which can be written (Platt and Stutz, 2007):

$$I(\lambda) = I_0(\lambda) \cdot exp(-\sigma(\lambda) \cdot c \cdot L) \,. \tag{1}$$

This law allows the definition of Optical Thickness ($\tau$) (Platt and Stutz, 2007):

$$\tau(\lambda) = ln\left(\frac{I_0(\lambda)}{I(\lambda)}\right) = \sigma(\lambda) \cdot c \cdot L \tag{2}$$

In a laboratory setting, Equation 1 or Equation 2 can be used to directly calculate an absorber's concentration, provided knowledge of the its cross section. In the open atmosphere however, absorption spectroscopy techniques are far more complex. On one hand, $I_0(\lambda)$ is not accessible, since we are measuring from inside the medium we want to measure. On the other hand, there are several environmental and instrumental effects that influence measurement results. These effects, such as extinction phenomena and instrumental light scattering, must be accounted for in a new equation (Platt and Stutz, 2007):

$$I(\lambda) = I_0(\lambda) \cdot A(\lambda, \ldots) \cdot \tag{3}$$
$$\cdot exp\left[ -\int \left[ \left(\sum_i \sigma_i(\lambda, s) \cdot c_i(s)\right) + \right.\right.$$
$$\left.\left. + \epsilon_M(\lambda, s) + \epsilon_R(\lambda, s) \right] ds \right]$$

Where:

$I(\lambda)$  is the light intensity as measured by the system;

$I_0(\lambda)$  is the intensity of the sun light before reaching Earth's atmosphere;

$A(\lambda)$  is the ratio of scattered light reaching the analysis point, accounting for the system's location and geometry.

$\sigma_i(\lambda, s)$  is the absorption cross section of absorber $i$;

$c_i$  is the concentration of absorber $i$;





$\epsilon_R(\lambda)$ represents Rayleigh's extinction coefficient;

$\epsilon_M(\lambda)$ represents Mie's extinction coefficient;

The interest of this equation lies within the retrieval of $c_i$, a given absorber's concentration. Since the integral is taken along the total atmospheric path of the measured photons, and considering that their cross sections do not vary significantly in atmospheric conditions, it is possible to define the concept of slant column, which is of great importance (Merlaud, 2013).

$$SC_i = \int_{atmosphere} c_i(s)ds \tag{4}$$

This quantity, as Equation 4 shows, equals the integral of an individual absorber's concentration along the atmospheric optical path of relevance. This is the quantity returned by passive DOAS measurements.

The introduction of the slant column and the integration of Mie and Rayleigh's coefficients to their optical depths, allow Equation 3 to be rewritten in the following way:

$$I(\lambda) = I_0(\lambda) \cdot A(\lambda, \ldots) \cdot \tag{5}$$
$$\cdot exp\left[ - \left( \sum_i (\sigma_i(\lambda) \cdot SC_i(\lambda)) + \tau_M(\lambda) + \tau_R(\lambda) \right) \right]$$

This is where the principle of DOAS is applied. Instrumental and atmospheric scattering effects have broad absorption spectral profiles. These can be separated from "quicker" and more important spectral signatures (Danckaert et al., 2015):

$$\sigma(\lambda) = \sigma'(\lambda) + \sigma_0(\lambda) \tag{6}$$

Therefore, the quick part of the optical density (called differential) can be isolated from the other contributions, which can be approximated and replaced by a low order polynomial.

$$ln\left(\frac{I_{ref}(\lambda)}{I(\lambda)}\right) = \sum_{i=1}^{n} \sigma_i'(\lambda) \cdot \Delta SC_i + \sum_{j=0}^{m} a_j \cdot \lambda^j \tag{7}$$

In Equation 7, $I_{ref}(\lambda)$ is the light intensities for a given scattered light spectrum, which is used as a reference because $I_0(\lambda)$, the light intensity outside Earth's atmosphere, is not accessible in typical DOAS experiments (satellite measurements are the exception); $\sigma_i'(\lambda)$ is the differential (the "faster") cross-sections and $\sum_{j=0}^{m} a_j \cdot \lambda^j$ a low order polynomial.

In practice, the mathematical solving of Equation 7 is not enough, since it does not account for the Ring effect or the non-linearities that result from stray light and wavelength shift in measured and cross-section spectra.

From the occurrence of these phenomena, it results that the mathematical procedure for DOAS measurements consists in solving a linear and a non-linear problem. The linear problem is solved by writing Equation 7 in its matrix form:

$$\tau = A \cdot X \tag{8}$$





In which $\mathbf{A}$ is an $m \times n$ matrix, with its columns being the differential cross-sections $\sigma_i^{'}(\lambda)$ and the wavelength powers taking the polynomial $P(\lambda) = \sum_{j=0}^{m} a_j \cdot \lambda^j$ into account. Since the number of lines in $A$ is much larger than the number of columns, the system is overdetermined. This means that there are many possible solutions and a principle must be defined in order to choose the best one. In this kind of problem, it is common to use the least-squares approach, in which the best solution

is the one that minimises $\chi^2 = [\tau - A \cdot X] \cdot [\tau - A \cdot X]^T$.

While the Ring effect is compensated through the use of a synthetically produced spectrum as an absorber, non-linearities are addressed by applying Levenberg-Marquardt's approach to non-linear fitting problems to equation (Merlaud, 2013; Bevington and Robinson, 2003):

$$ ln\Big( \frac{I_{ref}(\lambda)}{I(\lambda + shift) + offset} \Big) = \sum_{i=1}^{n} \sigma_i^{'}(\lambda) \cdot \Delta SC_i + \tag{9} $$

$$ + \sum_{j=0}^{m} a_j \cdot \lambda^j $$

Where $shift$ and $offset$, which represent spectral wavelength shifts and stray light offsets, respectively, are responsible for the non-linear character of the problem.

The FFF system and its algorithm are based on the Passive DOAS technology, by making scattered sunlight spectral measurements. However, the FFF algorithm is not a direct application of the said technique. For one, the system is continuously and

automatically measuring its surroundings and, on the other hand, it has a very particular goal - fire detection through smoke. Therefore, hardware and software must be adapted to the task at hand, as described in Section 4.

## 4  The Device

The Forest Fire Finder is a remote sensing system that has the goal of detecting forest fires. It is a sophisticated piece of equipment with many features and customisation possibilities. Its complexity and the fact that it is meant to operate 24 hours

per day create a need for control electronics and instrumentation. These devices are out of scope for this paper and will be revisited for another article that will include a detailed description of the FFF control software. This section aims to give a brief and basic hardware/software presentation for daytime spectroscopic operation and fire detection.

The FFF scans the horizon for the presence of a column of smoke, by performing sequential spectroscopic measurements of its surrounding environment using only the Sun as a light source, as illustrated by Figure 5. Sunlight is captured with a

Maksutov-Cassegrain telescope and guided through an fibre-optics cable into a spectrometer, which will transform it into an electric signal.

The system has to cover wide areas, which is why the telescope is mounted on an ENEO VPT501 Pan & Tilt Unit that ensures the device's movement. The pan & tilt head unit assembly also includes a FullHD camera, which is used primarily for the optical alignment of the system and for human validation and supervision. During the night, this camera is also used for

fire detection purposes, but that is not within the scope of this paper and shall also be approached in another publication.





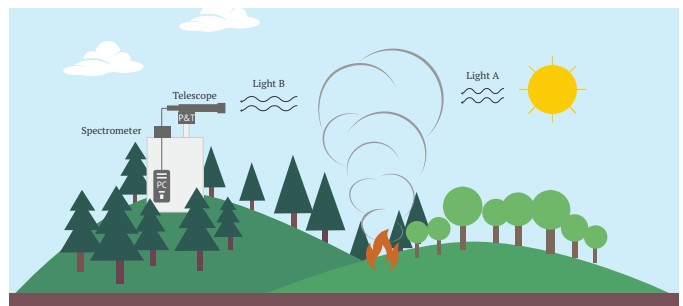

**Figure 5.** FFF Illustration. The system continuously scans the horizon, in search of a smoke column.

The Maksutov-Cassegrain telescope design uses the folded tube of the Cassegrain types and the spherical shape of primary mirror, secondary mirror and corrector lens of the Maksutov. In the FFF case, the chosen 90 mm aperture and 13.8 f ratio telescope represents the best compromise between size, magnification and amount of captured light. In addition, it is also a cost-effective solution for the task at hand.

The AvaSpec 2048 is a popular 2048 pixels CCD photo array spectrometer. It can be customised with several slit sizes and gratings in order to suit the application it is intended for. In the case of the FFF system, a 50 μm slit is used in conjunction with a 300 lines/mm grating, which ensures a wavelength range of 800 nm, from 300 nm to 1100 nm at a spectral resolution of 2.4 nm.

The spectrometer is connected to a computer, which is responsible for data processing and fire detection. It runs a custom
made software, developed in Mathworks' MATLAB development suite and C#. This software is deployed as a Microsoft Windows Service, as part of the FFF Software Suite.

## 5  Automatic Smoke Detection

The Forest Fire Finder is an electronic device that performs a spectroscopic analysis of the sky above the horizon, with the aim of detecting the presence of a smoke column. Smoke detection depends on the fire's emissions, which influence the composition
of the atmosphere and on the system's spectroscopic algorithms, which allow those changes to be detected.

### 5.1  Forest Fire Emissions and DOAS

Forest Fire smoke is a complex mixture of gases and aerosols that considerably changes the atmosphere (Urbanski et al., 2008). Among its key components are carbon oxides (CO and $CO_2$), Methane ($CH_4$), non-methane hydrocarbons, volatile organic components, nitrous oxides ($NO_x$), and particulate matter (Van Der Werf et al., 2010; Ward and Hardy, 1991). Trace
gases in smoke have a definite impact on the atmosphere's optical properties since some absorb light in the visible region of the electromagnetic spectrum. In addition to this, and depending on the combustion process, fire gives rise to the formation and emission of solid particles (Ward and Hardy, 1991). Given their size, these particles become aerosols, which influence light in





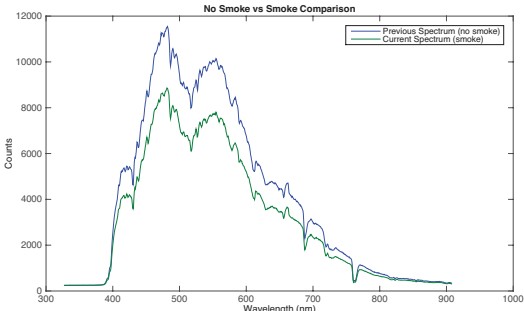

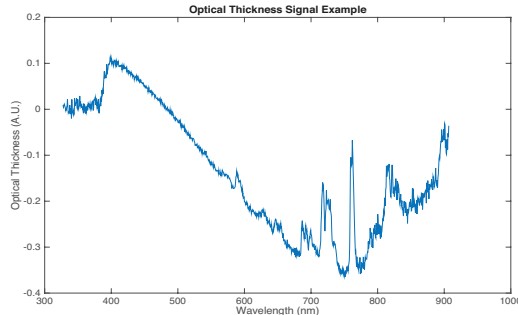

**Figure 6.** This plot shows how a smoke column can influence a spectral measurement. Both these spectra were acquired on the $29^{th}$ of December 2014, with a time difference of approximately 5 minutes and no azimuth difference.

**Figure 7.** Example of an Optical Thickness signal, obtaining by dividing two consecutive spectra of the same azimuth and calculating the logarithm of this division. It is this signal that is going to be fitted through Equation 9.

all wavelengths due to Mie's scattering. On the other hand, fire emissions also alter the balance between the perceived column densities of water ($H_2O$), oxygen ($O_2$), ozone ($O_3$), and of the oxygen dimer ($O_4$). All of these chemicals' cross sections are significant in the visible part of the spectrum.

Passive DOAS measurements are commonly used to retrieve the atmospheric column densities of several chemical com-
5 pounds. Smoke columns, however, present themselves as sudden and localised changes in atmospheric concentrations. If one were to use this technique and analyse their absolute concentration values *per se*, it would be very difficult to infer the presence of smoke. This does not mean passive DOAS cannot be used in this context. In fact, this method is very effective in detecting smoke if we put a "smokey" spectrum as $I$ and a "normal" spectrum as $I_0$ in Equation 2, resulting in a signal as displayed in Figure 7. Thus, by continuously acquiring spectra in a set of fixed azimuths and comparing the retrieved DOAS signals in
pairs by azimuth, a narrow and abrupt change such as the one produced by a forest fire becomes discernible in time. The FFF algorithm, presented in Section 5.2, does precisely this. These alterations are often difficult for the human eye to see, but there are some artificial intelligence algorithms that have been shown to be effective in separating the sky from a smoke column event and which we will discuss in Subsection 5.2.2.

### 5.2 The FFF Algorithm

The FFF algorithm spectroscopically detects the presence of a smoke column above the horizon. It relies on the system's continuous movement at constant speed to provide spatially accurate detections. The spectrometer acquires 2 spectra/s, which are all analysed by the computer. The algorithm can be divided into two phases: the chemical phase and the classification phase.




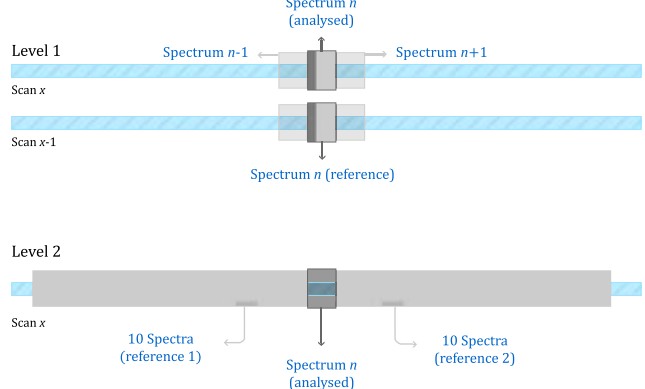

**Figure 8.** Illustration of FFF's two processing levels in the chemical stage of the algorithm.

### 5.2.1 The Chemical Phase

This algorithm section happens immediately after spectral acquisition. It corresponds to a passive DOAS analysis (see Section 3) of the spectrum in two different conceptual levels, as illustrated in Figure 8. The first level uses the same azimuth spectrum of the previous scan as a reference spectrum in the DOAS calculations. The second level uses the mean of the ten spectra immediately to the left and to the right of the analysed spectrum for the same purpose. This processing level was created in an empirical way, after observing that in the presence of strong winds, smoke columns move horizontally. Both levels are calculated using literature spectra, compiled in Table 1. In practice, these two processing levels represents three possible ways of applying Equation 2: considering $I_0$ as the previous spectra acquired with the same azimuth; considering $I_0$ as the average of the ten immediate spectra to the left of current azimuth; and considering $I_0$ as the average of the ten immediate spectra to the right of current azimuth (this last method implies delayed processing of the current spectrum).

In this stage, the algorithm uses an adapted Levenberg-Marquardt algorithm to calculate column density values for $NO_2$, $H_2O$, $O_2$, $O_3$ and $O_4$ through Equation 9. In parallel, short and long wavelength energy contribution and Signal to Noise Ratio (SNR) are also computed for the analysed spectrum optical density. All the processed data are stored in a single matrix, which will be used by the second stage of the FFF algorithm.

### 5.2.2 The Classification Phase

This stage runs at the end of each scan, processing chemical data from each spectrum in sequential order. A work-flow diagram is presented in Figure 9. The process begins by selecting the spectra that need to be further analysed. This is done by assembling a signal composed of the average signal energy per pixel of each spectrum and by running it through a peak detection routine.

The algorithm then proceeds by calculating column density ratios between $O_2$, $O_4$, $NO_2$, $H_2O$ and $O_3$ for the spectra of the detected mean energy peaks. In the next step, the program compares the SNR of the analysed spectrum (see Subsection 5.2.1)





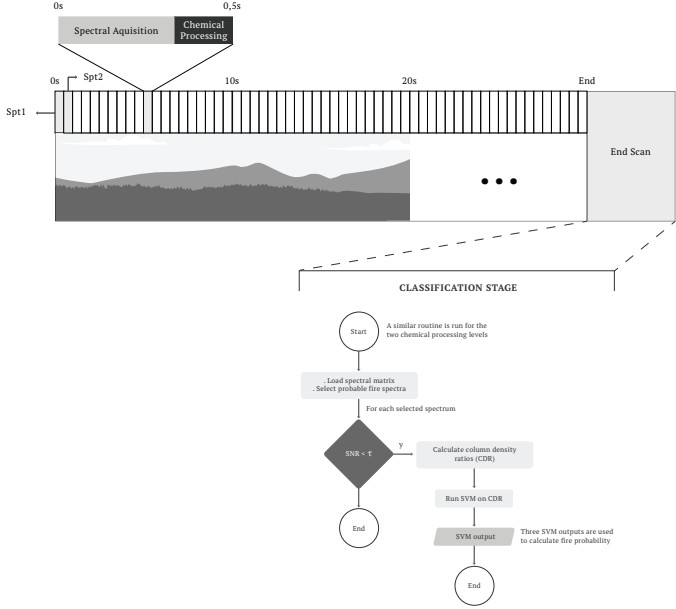

**Figure 9.** FFF Algorithm simplified work-flow diagram.

with a fixed threshold set in a configuration file. If this value is deemed acceptable, calculated column density ratios are entered into a Support Vector Machine (SVM) as inputs.

An SVM is an algorithmic approach to the problem of classification in the context of supervised learning (Press et al., 2007). Introduced in 1992, by Boser et al. (Boser et al., 1992), this method has since proved itself of great usefulness by providing relatively straightforward solutions to previously complicated classification applications. SVMs are generally easier to implement and understand, and this has also contributed for their fast widespread. The general concept behind the SVM methodology is to find and define the hyperplane that better separates data in two classes (Press et al., 2007).

Like all supervised learning techniques, SVMs need to be trained prior to being used. In the case of the FFF classification algorithm, SVMs were trained in successive generations, with each generation built upon the results of the previous.

The first generation SVM was trained using data from 60 different moments in 2014. Of these data, 30 moments corresponded to a fire, while the other 30 corresponded to clouds that looked visually similar to smoke columns, for a trained observer. Although not ideal, this is a valid first approach because the system's sensitivity is limited to the visible part of the electromagnetic spectrum.

The two chemical processing levels originate three classification results:

– Smoke column detected between previous and current scan;

– Smoke column detected on the left of analysed spectrum;

– Smoke column detected on the right of analysed spectrum;



**Table 1.** Literature spectra used for the FFF's passive DOAS calculations. All cross-sections are downloaded from the MPI-Mainz UV/VIS Spectral Atlas of Gaseous Molecules of Atmospheric Interest (Keller-Rudek et al., 2013)

| Compound | Reference Key | Year |
|---|---|---|
| Oxygen ($O_2$) | Bogumil et al. (2003) | 2003 |
| Ozone ($O_3$) | Bogumil et al. (2003) | 2003 |
| Oxygen Dimmer ($O_4$) | Hermans (2011) | 2011 |
| Water Vapour ($H_2O$) | Coheur et al. (2002) | 2002 |
| Nitrogen Dioxide ($NO_2$) | Vandaele (2002) | 2002 |

If two of these classification results are positive, the system issues an alarm.

# 6 Results and Discussion

In 2013, 13 FFF devices were deployed in the Peneda-Gerês National Park (PNPG), in the north of Portugal. Their placement reflected topography, local accessibility and fire protection needs.

In 2015, FFF data was gathered and compared to official data from the Portuguese National Authority of Civil Protection (ANPC), the country's responsible institution for forest fire protection and management.

Table 2 shows said data and comparison. In it, a confirmed detection occurs when a smoke column is sensed by an FFF device and the detection is validated by a human operator. This is different from a Registered Fire Event (RFE), which is a fire that was inserted into ANPC's database.

Official statistics count 132 fire events in 2015 within PNPG. During the same period, the FFF network issued 578 detections, of which 369 were false detections and 209 confirmed events, of which 53 were coincident with RFEs.

Although the false detections may seem to be high in comparison to confirmed detection, it is important to bear in mind that each system has an average working period of 12 hours per day. At two spectra per second, this means an average of 86400 analysed spectra per system per day. Since each and every one of these analysis can trigger an alarm by itself, false detections reach only 0.0000833 % in all systems. In addition, there have been some events that were wrongly marked as false detections due to misunderstandings on how the system is to be handled by humans. These events correspond mainly to small fires and prescribed burns, large enough to be detected. Security issues, regarding the Portuguese Civil Protection Authority, prevent the exact quantification of human errors, but they amount to a significant percentage of false alarms.

Although the presented numbers are enough to paint a general picture of the FFF system's behaviour, the available data does not allow a thorough quantification of the system's performance, since there is no correct gold standard regarding forest fires due to fire registry procedures not being clearly established. This becomes exceedingly apparent when comparing the number



**Table 2.** FFF Statistics for 2015 in the Peneda-Gerês National Park.

| | |
|---|---|
| **Registered Fire Events** | 132 |
| **Total Fire Detections** | 578 |
| **False Events** | 369 |
| **Confirmed Detections** | 209 |
| **Estimated Network Operation Time** (h) | 56940 |
| **Estimated Analysis** | 409968000 |
| **False Detection** % | 0.0000900 |
| **False Alarms per System/Day** | 0.07776607 |

of RFEs and the number of confirmed detections: every confirmed detection was deemed relevant by a human operator, yet there are only 132 RFEs for the 209 confirmed detections.

Another important result that becomes noticeable in Table 2 is the fact that false positives and true detections vary in similar ways. This can be explained by the fact that the FFF is a spectroscopic system at its heart. Fire releases chemical components into the atmosphere, which in turn are detected by the system. If there are many fires in a small geographic region, such as the Peneda-Gerês park, it is possible that an FFF is able to sense it, without the presence of a visible smoke column in its patrol path. We have also noticed a trend for false alarms in specific cloudy days. We believe this is due to pollutant particles carried by the clouds. Light scattering by these particular clouds sometimes breaches through the system's energy and SNR filters (see Subsection 5.2.1) and is incorrectly classified as a smoke column.

## 7   Future Developments

The FFF has been an ongoing development for NGNS-IS, Ltd.. Since 2006, the device has undergone two complete re-design processes, motivated by hardware improvements.The current version is without a doubt the most robust and reliable design ever, achieving uptimes of 99%. There will always be room for improvement regarding hardware, but given the operating level of the current version, these will not be a priority in the near future.

The software architecture selected for the system allows total freedom for future development needs, with minimal integration efforts. As stated in Section 4, this was a requirement because most customers need some level of customisation.

A new version of spectral algorithms is currently in development, with an estimated release date of 6 months. This version will take into account spectral stretch in the fitting procedure, instead of only considering spectral shifts. It is intended that this new version will improve false alarm rates (see Section 6) and decrease processing time (see Section 5).



DOAS (see Section 3) is a widely used atmospheric analysis technique, with much broader uses than just fire sensing. The experience attained while developing FFF allowed the creation of another project, called Project ATMOS, designed to monitor and control air pollution, crops maturity and water stress, and agricultural plagues. This project started in April 2016 and is expected to conclude in September 2019.

## 8  Conclusions

Life on Earth is greatly influenced and shaped by fire events. Humans in particular depend on fire to maintain their technology and way of life. However, forest fires are a global menace that cause concern all over the world. Several industrialised countries have allocated a great deal of resources to researching wildfires and their behavioural dynamics.

It is this concern that generates a very large market for remote sensing equipment for early forest fire detection, a market which is expected to grow 11.53% until 2020.

This article addresses one of such equipments. The Forest Fire Finder, or FFF, was developed in Lisbon in 2006 by NGNS-IS, Ltd. and is the only one that is based on optical spectroscopy, particularly Differential Optical Absorption Spectroscopy.

In 2013, a 14 element network of FFF devices was installed in the Peneda-Gerês National Park. In 2015, this network was able to detect a confirmed 209 fire events, a number significantly higher than the officially registered 132 fire occurrences in the same region.

For the same period the system has issued 369 false detections, but it is worth considering that confirmed and false detections have similar trends. This is due to the fact that fires pollute the atmosphere with the chemicals that the system aims to detect, and is in agreement with the device's operating physical principle, optical spectroscopy.

Evaluation of a fire detection system is a very difficult task. There is no formal definition of how large a forest fire must be to be considered an event and this means there is no perfect classification model (a gold standard) to compare the system's performance to.

The FFF is an automatic forest fire detection system which has proven itself to be effective and detected a great number of forest fires (see Table 2 in Section 6). In addition, the system's current hardware and software configuration has resulted in extremely high uptime levels, contributing for an adequate fire detection coverage and consequently, optimal levels of fire protection.

*Acknowledgements.* The authors would like to thank the NGNS team for their support, especially Hélia Pinto, for the numerous drawings and diagrams.





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
