# Peer review of "Forest Fire Finder - DOAS application to long range forest fire detection."

_Atmospheric Measurement Techniques, 2016_

## Referee Comment (RC1) · Anonymous Referee #1 · 12 Dec 2016

The paper by Valente de Almeida and Vieira presents a detection system for forest fires based on the DOAS technique, and examples of its operation in 2015 in Portugal. The concept is interesting and should be published, nevertheless, there is room for improvement before publication.

Major Comments

The paper is too concise and does not present accurately enough the technical details. Lots of space is used to describe basic concepts (for instance of the DOAS technique) so the information content on what is actually new in the study is too sparse. The authors seem to have done enough and collected enough data for a paper, but in my opinion they need to explain more what they do.

Figure 1 and Figure 3 are not very relevant for the study since they describe set-up which are not used (wireless sensor network, active doas). However, we miss a figure with a map of the 13 FFF system in the forest. We also miss typical DOAS fits corresponding to a fire detection, and a picture of the instrument on site. As described, the detection is a black box to the reader. We read that the system is trained to recognize smoke scenes, but what are the main criterion for the presence of smoke in the algo in the end?

The authors mention twice that other papers are in preparation: P7 l 21 'These devices are out of scope for this paper ... will be revisited for another article...'. and P7 L 30: 'During the night, this camera is also used... shall also be approached in another publication'. As the paper is rather short, I believe this additional material could be added to the paper instead of adding publication.

Minor Comments

Introduction

The authors should mention the species emitted by fire (NOX, CO ...) and include in the reference list some DOAS studies on forest fire, eg from space

Spichtinger, N., Damoah, R., Eckhardt, S., Forster, C., James, P., Beirle, S., . . . Novelli, P. C. (2004). Physics Boreal forest fires in 1997 and 1998 : a seasonal comparison using transport model simulations and measurement data, 1857–1868.

Castellanos, P., Boersma, K. F., and van der Werf, G. R.: Satellite observations indicate substantial spatiotemporal variability in biomass burning NOx emission factors for South America, Atmos. Chem. Phys., 14, 3929-3943, doi:10.5194/acp-14-3929-2014, 2014.

I also suggest to mention other routine monitoring applications using DOAS such as volcanic monitoring from space

Brenot, H., Theys, N., Clarisse, L., Geffen, J. Van, Gent, J. Van, Roozendael, M.

Van, & Hurtmans, D. (2014). Support to Aviation Control Service ( SACS ): an on-line service for near-real-time satellite monitoring of volcanic plumes, 1099–1123. http://doi.org/10.5194/nhess-14-1099-2014

or from ground

Galle, B., Johansson, M., Rivera, C., Zhang, Y., Kihlman, M., Kern, C., ... Hidalgo, S. (2010). Network for Observation of Volcanic and Atmospheric Change (NOVAC)—A global network for volcanic gas monitoring: Network layout and instrument description. Retrieved from http://www.agu.org/pubs/crossref/2010/2009JD011823.shtml

The author could quote IPCC and/or references therein for the observed and predicted increases in number of forest fires.

On the other hand, the prediction of 11.53% increase of the market between 2014 and 202O (p2 l.11) does not seem serious.

State of the art.

References are needed for the satellite instruments

3. The technique

Active DOAS could be skipped, both text and figure since it is not used in the study

If the authors choose to be pedagogical with DOAS, they need to explain the symbol used in the equations, this is done after eq 3 but not eq 1 and 2 so the explanation of I Io, sigma, c and L should be at the beginning for the sake of clarity

Equation 4 should be modified, the integral is on the optical path for the slant column, the authors wrote that the integral is on the atmosphere which can be confusing with the vertical column.

P 6 L 22 We miss a reference for the Ring effect

P 7 L 3 Overdetermined does not imply that there are many possible solutions, on the

contrary, but it does imply a criterion to select a state close to the solution.

The authors seem to have used the QDOAS software but it s not explicit. Can they be more explicit on that?

Figure 7: The y axis for Optical thickness is labeled 'AU' but the optical thickness is a dimension less quantity

Section 5.2.2

As mentioned above, we miss details and figures here. The authors should also be more explicit on the training of their algorithm. Where were recorded the spectra? At the same place, with fires? What are the columns of the different absorbers in the 'detected' spectra?

Section 6

It s not clear to me how we can say that true detection and false positive vary in similar way. I understand the authors'explaination of why it could be the case but how can we see it from the table 2? It seems that we would need time series to see such a link.

About the false detection on cloudy days: with more details on the different columns for the 'detected' case, the authors could discuss more accurately their assumption. These false alarm could be linked with variation in the O4 column due to clouds, which may also occur in fire smoke.

Technical comments

P.5, l.6 : 'of (the) its cross section'

P.7 L.25 : 'an fibre otpics' -> an optical fiber

Conclusion

Again the 11.53% which is hard to believe.

---

## Referee Comment (RC2) · Anonymous Referee #2 · 19 Dec 2016

This paper describes the use of a DOAS system for the detection of forest fires.

This system is a very interesting application of DOAS technique for the indirect detection of the smoke columns over the horizon.

However, the explanation of the physical principle that resides under the indirect detection of the smoke using DOAS, that is, what can DOAS detect that can be understandable as smoke should be much more detailed in the text.

The article should address some mayor and minor comments prior publication.

Mayor comments

In this paper is very confusing not to know what is going to be retrieved by using something that is based in DOAS but is not exactly DOAS. In the section "Automatic smoke detection" it is mentioned that the column of several absorbers as water vapour, $O_2$, $O_4$ and $O_3$ shows an abrupt change during a forest fire event. Lately $NO_2$ is also mentioned as a target specie. Please unify criterion.

Authors should explain in more detail what the target species for identification of presence of smoke are and why (for example showing a figure of the mentioned abrupt variation). Molecular oxygen is not a usual target for DOAS retrieval, please could authors indicate in what spectral interval is retrieved?

Even when the exact calculation of SCD of $O_2$, ozone, $NO_2$ or $O_4$ SCD is not an objective itself for FFF, it would be necessary to see a figure with the spectral fits of these species without and with smoke, in order to see how well performs the DOAS part of this instrument. A more detail in the parameters used in the analysis would be also necessary (i.e., interval fit for every absorber). Temperature dependence of $O_3$ and $NO_2$ absorption cross sections is not taken into account even when temperature of the scanned air mass is an important feature for this application, could authors explain why?

What is the reason for the mentioned strong variation observed in the SCD when smoke column is detected? Change in concentration, change in optical path, change in the temperature on the air masses over the forest fire? Strong variation observed into the retrieved SCD for some gases, should be better documented and I would like to see a figure where this abrupt change could be seen, retrieved using the FFF system.

It is necessary at this point to know what is the reason that provokes this strong change in the observed SCD, in order to detect the reason for false positives: how can be, for instance, a situation during a Saharan dust event distinguished from the presence of a smoke column?

The detection of smoke using FFF is also based in the observational strategy, SCD of absorbers is calculated by using a set of three different reference spectrum. This should be explained in more detail, an example of the selection of the spectrum that is selected for further analysis, as mentioned in page 10, line 28 would be very helpful. It is not very clear to me why is necessary to compose an average spectrum and running through a peak detection routine. Why is necessary to detect peaks in the spectrum? It not clear either what average the system is calculating: what spectra are used? How this can lead to isolate the target spectrum?

I found that there is a connection between the three kind of references mentioned in page 10 section 5.2.1 and the three classification results enumerated in page 11 section 5.2.2., but it is no easy to link these two facts from the text, please explain this in more detail.

Minor comments.

Please make larger figures, some of them as figure 5, 6, 7, 8 and 9 have very small fonts and are difficult to read.

In the State of the art, page 2, three different kind of methods for forest fire detection are mentioned. It would be interesting to address in which of the three methods can be FFF included (if any) or indicate if FFF relies on a different new method that has been not used at the moment. It is no clear to me if FFF could be addressed into the "Large Area Remote Sensing".

Figure 1 is not significant for this work and should be removed.

Page 3, line 8. Please convert miles into km.

In the part corresponding to the brief description of the commercial devices used at the date for forest detection, it would be useful to know what the advantages of FFF over the previous systems are (minimal human intervention, maybe cost?).

It is also mentioned in this part of the text that as FFF operates in an outdoor scenario, results are not as quick as they could be in a lab experiment. This is the case for most monitoring DOAS systems, but I imagine that in this kind of application, results should be available in real time, is this the case of this system? This should be mentioned in the text.

In the section "The technique", I found the explanation a little bit messed up. For instance, magnitudes on the equation 1 are not explained until 10 lines below (and not all of them). They should be explained next to the equation, especially when in line 5-9, there is some discussion on $I_0(\lambda)$ when it has not been defined yet. Please re-organize this section.

Page 5 line 8. Amongst the environmental effects that affect DOAS measurements, multiple scattering should be included as the aim is detect smoke.

Page 5 line 9. Please explain what is instrumental light scattering, are authors referring to straylight here?

Equation (3). Please explain $A(\lambda,…)$ in more detail. What "…" is? In line 16 page 3 is defined as a ratio but it is not indicated of what magnitudes is this ratio.

Page 6 line 8. $SC_i$ is mentioned as the result of passive DOAS measurements, but this is only true when $I_0((\lambda)$ is known, what is not usually the case for this kind of measurements. In fact this erroneous concept is propagated along the explanation affecting other parts of the work. This connect to figure 7, how can authors explain the existence of negative optical thickness? (please notice as well that optical thickness is a non-dimensional magnitude).

Page 6 line 14,

by spectral profiles authors mean spectral features?

"quicker" and more important spectral signatures, is not a clear nomenclature, please rewrite this part.

Equation (6) magnitudes are not defined.

Equation (7) needs further explanation. Where the magnitude $A(\lambda,...)$, Mie and Rayleigh scattering terms are?. What the new terms $\Delta SC_i$ and $a_j\lambda^j$ are? To eliminate $I_0(\lambda)$ is not as straightforward as authors mean, please notice that Iref is also affected by absorption and scattering and this is not mentioned in the step from equation (5) to (7), this is a part of what I meant in the previous comment of line 8 in this same page.

Page 7 line 6, Ring effect is not compensated but treat as an absorber by using a pseudo-absorption cross section. A small explanation, as this part of the work is mainly pedagogical, should be included about the calculation of this cross section and please include a reference.

Page 7 line 14. Please, if FFF does not apply directly DOAS technique itself, please indicate the differences with DOAS or in which way the technique is going to be used. This is an important part, especially when an important effort has been made to introduce DOAS technique.

 Page 7 line 15. System is not measuring its surroundings but something in its surroundings, please specify what.

Section "The Device" What is the Field of View of the telescope?

Section "Results and Discussion"

I understand that the percentage of false positives are low when compared to the high number of analysed spectra per day, but the reliability of this system should reside on the correct detection of forest fires, so the analysis of false positives is an important part that need to be improved. Authors should indicate if further work is going to be done in this line and indicate how this weakness of the system can be improved.

It is interesting that most of false positives are due to presence of clouds, especially when DOAS can be also used to detect clouds and even aerosols. I don't know if authors are aware of the previous work in this area (Gielen et al., Atmos. Meas. Tech., 7, 3509–3527, 2014 www.atmos-meas-tech.net/7/3509/2014/ or , Wagner et al., Atmos. Meas. Tech., 7, 1289–1320, 2014 doi:10.5194/amt-7-1289-2014).

---

## Author Comment (AC1) · 20 Dec 2016

We begin by saying that we have great pleasure in reading and discussing comments to our work and by thanking the referee for the comments.

We will divide this document in order to address the points raised by the referee individually and in order, as they were presented to us.

**Major Comments**

**Comment:**

*'Figure 1 and Figure 3 are not very relevant for the study since they describe set-up*

*which are not used (wireless sensor network, active doas). However, we miss a figure with a map of the 13 FFF system in the forest. We also miss typical DOAS fits corresponding to a fire detection, and a picture of the instrument on site. As described, the detection is a black box to the reader. We read that the system is trained to recognize smoke scenes, but what are the main criterion for the presence of smoke in the algo in the end?'*

**Reply:**
Indeed, Figures 1 and 3 are not *directly* relevant to the study at hand, but we do feel they add a measure of depth to the subject by illustrating alternatives to the proposed methods. However, we acknowledge that they are not crucial for the study and will be happy to remove them if the referee maintains the stated opinion.

The 13 systems that were deployed in the Peneda Gerês National Park are property of the Portuguese National Authority of Civil Protection. The systems' locations are not publicly known and we are not authorised to disclose them. We can, nevertheless, include a picture of one of our systems.

Regarding the issue of detection, we acknowledge the referee's comment that it may seem to be a black box for the reader. However, we are not confident that we can make it transparent, starting with the DOAS typical fits for a fire detection. The device's operating mode means it compare spectra in order to find *relative* column densities. This in turn has two consequences:

- The device never knows absolute column densities or tries to calculate them;

- Human eyes are not able to differentiate between a fire and a non fire spectrum by looking at a fit.

As stated in the first paragraph of Section 5.2.2 of our article, we calculate column density ratio values for five chemical compounds in the atmosphere and then feed

these ratios into our Support Vector Machine (SVM). These algorithmic tools are used to find patterns that might indicate the presence of a smoke column. In this case, a five dimension problem, it would be very hard for a human to detect and identify these recurring patterns.

Now, SVMs, as all supervised learning techniques, demand that a training operation is performed prior to use. It is possible that this is the part that the referee felt lacking. We agree that the training process might be better explained. Nevertheless, after the training procedure and during the SVM operation, we do not possess information on how the separation takes place. We rely on the algorithm's proven classification validity.

**Comment:**
*The authors mention twice that other papers are in preparation: P7 l 21 'These devices are out of scope for this paper ... will be revisited for another article...'. and P7 L30: 'During the night, this camera is also used... shall also be approached in another publication'. As the paper is rather short, I believe this additional material could be added to the paper instead of adding publication.*

**Reply:**
Although we understand the referee's concerns over the article's conciseness, the topics which are said to be out of scope are really out of scope. On the one hand, they range from instrument and software design; from technical drawings to PCB design, through to the software architecture and implementation. On the other hand, they include a series of image processing routines, with no spectroscopic measurement of any kind, developed solely for night fire detection for this particular device.

**Minor Comments**

**Comment - Introduction 1:**

*The authors should mention the species emitted by fire (NOX, CO ...) and include in the reference list some DOAS studies on forest fire, eg from space*
*...*
*I also suggest to mention other routine monitoring applications using DOAS such as volcanic monitoring from space*

**Reply:**
We appreciate these suggestions and will take them into account.

**Comment - Introduction 2:**
*On the other hand, the prediction of 11.53% increase of the market between 2014 and 202O (p2 l.11) does not seem serious.*

**Reply:**
We have based this section of the text on a market research report by Research and Markets (www.researchandmarkets.com). Although a prediction is by definition not certain, we have no reason to doubt this company's methods nor to say that their presented numbers are in any way dishonest.

**Comment - State of The Art:**
*References are needed for the satellite instruments*
**Reply:**
We will complete our references for this section.

**Comment - Technique:**
*If the authors choose to be pedagogical with DOAS, they need to explain the symbol used in the equations, this is done after eq 3 but not eq 1 and 2 so the explanation of I Io, sigma, c and L should be at the beginning for the sake of clarity. Equation 4*

*should be modified, the integral is on the optical path for the slant column, the authors wrote that the integral is on the atmosphere which can be confusing with the vertical column. P 6 L 22 We miss a reference for the Ring effect. P 7 L 3 Overdetermined does not imply that there are many possible solutions, on the contrary, but it does imply a criterion to select a state close to the solution.*

**Reply:**
These are very valid points and we will address them accordingly.

**Comment - Technique 2:**
*The authors seem to have used the QDOAS software but it s not explicit. Can they be more explicit on that?*

**Reply:**
The QDOAS software was not used for the development of this article or this device. As we state in the last paragraph of Section 4, the system uses MATLAB and C# custom made routines. We have, however, used the software's manual as reference in Section 3.

**Comment - Technique 3:**
*Figure 7: The y axis for Optical thickness is labeled 'AU' but the optical thickness is a dimension less quantity*

**Reply:**
We appreciate the comment and will address the issue.

[Figure]

**Comment - Section 6 1:**

*It's not clear to me how we can say that true detection and false positive vary in similar way. I understand the authors' explaination of why it could be the case but how can we see it from the table 2? It seems that we would need time series to see such a link.*

**Reply:**

This is a valid comment. Table 2 does not show that information in any way and the reference should not be there. We will change this paragraph accordingly.

**Comment - Section 6 2:**

*About the false detection on cloudy days: with more details on the different columns for the 'detected' case, the authors could discuss more accurately their assumption. These false alarm could be linked with variation in the O4 column due to clouds, which may also occur in fire smoke.*

**Reply:**

As mentioned above, a detection does not present itself by a clear column pattern that can be identified by a human. Thus the need for artificial intelligence algorithms like the SVM.

As to the clouds and the O4 column variation, we are inclined to believe that while this can definitely be a contributing factor for a false detection (this is actually one of our current lines of research, though still in the early stages), but it certainly is not the only one. If that were the case, we would expect a much higher number of false alarms, given the discrepancy between the number of clouds and the number of fire events.

We hope we have given satisfactory answers to every point raised and remain ready to provide further information, should the need for it arise.

---

## Author Comment (AC2) · 10 Jan 2017

We begin by saying that we have great pleasure in reading and discussing comments to our work and by thanking the referee for the comments.

We will divide this document in order to address the points raised by the referee individually and in order, as they were presented to us.
* * *
**Major Comments**

**Comment:**

*'In this paper is very confusing not to know what is going to be retrieved by using something that is based in DOAS but is not exactly DOAS. In the section automatic smoke detection it is mentioned that the column of several absorbers as water vapour, O2, O4 and O3 shows an abrupt change during a forest fire event. Lately NO2 is also mentioned as a target specie. Please unify criterion.'*

**Reply:**

The Forest Fire Finder (FFF) system uses the mathematical ingenuity behind DOAS to try to detect a smoke column in the atmosphere and above the line of the horizon. Indeed, and as the referee has pointed out in several occasions throughout the comments, the FFF would be severely lacking in many different ways were it a pure DOAS system meant for quantitative atmospheric analysis. This is not the case, however.

Most of the device's "DOAS faults" can be explained by the time and memory constraints the system faces. As stated in Section 5.2, the FFF makes 2 acquisitions per second. This means that the chemical stage of the algorithm must ideally be performed in less than 500 ms minus the acquisition time (which dynamically varies from 60 to 450 ms and is typically around 210 ms). Our software calculations currently take between 250 and 350 ms, meaning that the software is almost always lagging behind acquisition. The system is designed to cope with this and does it with great robustness, but it does mean that at the end of any given scan, there is a time interval in which the system's computer is literally catching up to the scan. In a large scan (the scan has a maximum length of 900 spectra), this interval has a huge impact on the scan time. As mentioned in the paper's Section 7, decreasing processing time is one of our software development goals.

As to the omission of $NO_2$, it was a simple mistake, we thank the referee for pointing it out.

**Comment:**

*'Authors should explain in more detail what the target species for identification of presence of smoke are and why (for example showing a figure of the mentioned abrupt variation). Molecular oxygen is not a usual target for DOAS retrieval, please could authors indicate in what spectral interval is retrieved?'*

**Reply:**
We have retrieved a list of relevant trace gases of forest fires from several sources listed on our references. After that, we had to adapt that list to the optical capabilities of the equipment to which we had access, and we ended with two chemical components: $NO_2$ and $O_3$. Since fires consume oxygen and evaporate water in plants, we have empirically added $H_2O$ in the 400 -500 nm region and $O_2$ in the 600 - 800 nm region. The introduction of both these molecules greatly improved the classification, presumably also because it has improved the fitting quality. Finally, we have decided to introduce the oxygen dimer, $O_4$, because of its correlation with $O_2$ and the fact that it has a very broad cross section, with relevant structures all over the visible spectrum.

**Comment:**
*'Even when the exact calculation of SCD of O2, ozone, NO2 or O4 SCD is not an objective itself for FFF, it would be necessary to see a figure with the spectral fits of these species without and with smoke, in order to see how well performs the DOAS part of this instrument. A more detail in the parameters used in the analysis would be also necessary (i.e., interval fit for every absorber). Temperature dependence of O3 and NO2 absorption cross sections is not taken into account even when temperature of the scanned air mass is an important feature for this application, could authors explain why?*
*What is the reason for the mentioned strong variation observed in the SCD when smoke column is detected? Change in concentration, change in optical path, change in the temperature on the air masses over the forest fire? Strong variation observed into the retrieved SCD for some gases, should be better documented and I would like to see a figure where this abrupt change could be seen, retrieved using the FFF system.'*

**Reply:**
There seems to be a slight misunderstanding on behalf of the referee towards the classification stage of our algorithm, probably due to a poor choice of words on our part. Indeed the paper states that a forest fire's smoke represents an abrupt change, but in the same paragraph we also mention that this change is difficult for the human eye to see. Although sudden and confined to one or two spectra (remember the system is continuously acquiring spectra), the variation is very weak and is most often indiscernible from a *normal* observation, especially if there are clouds in the analysed portion of the sky.

The fact that these differences are so small is precisely the reason why we have resorted to using complex artificial intelligence algorithms such as the Support Vector Machine. The SVM is a supervised machine learning algorithm, which therefore has to be trained ahead of application. The learning process is initiated by manually separating a number of fire occurrences (30 for each successive generation of SVM, in our case) from a sample of non-fire analysis. This is possible for a trained technician using the photographic data which is also acquired by the system and by using a special software tool which was designed precisely for that operation.

The trained SVM recognises patterns in the data and tries to separate a smoky spectrum from a non smoky spectrum by fitting a plane between the two classes. In our case, and since we work in 5 dimensions, this is a hyperplane. While this makes it impossible to confirm where the pattern being recognised appears and what causes it, the fact that the SVM algorithm is mathematically valid allows us to confidently rely on the obtained results.

We will, nevertheless, reformulate this section of the paper, since it appears we could be quite clearer.

**Comment:**
*'...how can be, for instance, a situation during a Saharan dust event distinguished from*

*the presence of a smoke column?'*

**Reply:**
We have found this comment to be anecdotally curious. In fact, in the year 2013, there was such an event in Portugal as a Saharan dust cloud. The first generation of the FFF algorithms had severe difficulties during that period. It was based on the comparison of the optical density between two spectra and on energy contribution in various intervals and had a less sophisticated classification algorithm, based on a series of "if clauses". In the current system, it is unlikely that this Saharan dust would produce SCD acquisitions with sufficiently high Signal to Noise Ratios. If it did, it would also have to have a specific mean energy per pixel in order to pass through to the SVM. In this very unlikely scenario, the hypothetical Saharan dust SCDs would have to fit the exact pattern for which the classifier is looking for in order for it to be classified as fire. This is very unlikely, but since the system performs near 60.000 analysis per day, one would be wise to expect an increased probability of false alarms during this period.

**Comment:**
*'The detection of smoke using FFF is also based in the observational strategy, SCD of absorbers is calculated by using a set of three different reference spectrum. This should be explained in more detail, an example of the selection of the spectrum that is selected for further analysis, as mentioned in page 10, line 28 would be very helpful. It is not very clear to me why is necessary to compose an average spectrum and running through a peak detection routine. Why is necessary to detect peaks in the spectrum? It not clear either what average the system is calculating: what spectra are used? How this can lead to isolate the target spectrum? I found that there is a connection between the three kind of references mentioned in page 10 section 5.2.1 and the three classification results enumerated in page 11 section 5.2.2., but it is no easy to link these two facts from the text, please explain this in more detail.'*

**Reply:**
We have read this comment thoroughly and with a great deal of attention, but we have

to say we are not in concordance with the referee. Page 10 of the FFF article introduces the chemical phase of the algorithm used to detect a smoke column. This section of the algorithm uses two levels of processing which we explain in the first lines of the first paragraph:

> "...
> The first level uses the same azimuth spectrum of the previous scan as a reference spectrum in the DOAS calculations. The second level uses the mean of the ten spectra immediately to the left and to the right of the analysed spectrum for the same purpose. ..."

In addition, the quoted text is complemented with Figure 8, which clearly illustrates both levels of processing.

As to the second part of the referee's comment, which addresses the beginning of the selection process, we believe that the referee may have misread. In this subsection, we do not mention (nor does the system calculate) an average spectrum. In the article, we state:

> "The process begins by selecting the spectra that need to be further analysed. This is done by assembling a signal composed of the average signal energy per pixel of each spectrum and by running it through a peak detection routine."

It is this assembled energy signal, gathered from all the spectra in the scan, that is the target for the peak detection routine.

The last part of this comment is directed at the last part of the classification subsection, where the referee felt there should be a mention to the three references used in the

chemical stage. We also disagree with this comment, as we do make the mention the referee felt lacking in lines 14 to 17 of page 11. We will try to improve the link between subsections, so that this reference becomes more apparent.

**Minor Comments Comment:**
*'Please make larger figures, some of them as figure 5, 6, 7, 8 and 9 have very small fonts and are difficult to read.'*

**Reply:**
We appreciate this comment and will take it into consideration.

**Comment:**
*'In the State of the art, page 2, three different kind of methods for forest fire detection are mentioned. It would be interesting to address in which of the three methods can be FFF included (if any) or indicate if FFF relies on a different new method that has been not used at the moment. It is no clear to me if FFF could be addressed into the "Large Area Remote Sensing".'*

**Reply:**
The description of the "Large Area Remote Sensing" device family continues into page 3. In the first line, one can read:

> "There are already several commercially available systems [in the Large Area Remote Sensing family], such as the Forest Fire Finder..."

**Comment:**
*'Figure 1 is not significant for this work and should be removed.'*

**Reply:**
We do feel the figure adds depth to the subject, but since this has been observed twice, we are happy to remove it.
**Comment:**

*'Page 3, line 8. Please convert miles into km.'*

**Reply:**

We appreciate this comment and will take it into consideration.

**Comment:**

*'In the part corresponding to the brief description of the commercial devices used at the date for forest detection, it would be useful to know what the advantages of FFF over the previous systems are (minimal human intervention, maybe cost?). It is also mentioned in this part of the text that as FFF operates in an outdoor scenario, results are not as quick as they could be in a lab experiment. This is the case for most monitoring DOAS systems, but I imagine that in this kind of application, results should be available in real time, is this the case of this system? This should be mentioned in the text.'*

**Reply:**

These are indeed valid points. We will address them accordingly.

**Comment:**

*'In the section "The technique", I found the explanation a little bit messed up. For instance, magnitudes on the equation 1 are not explained until 10 lines below (and not all of them). They should be explained next to the equation, especially when in line 5-9, there is some discussion on $I_0(\lambda)$ when it has not been defined yet. Please re-organize this section. Page 5 line 8. Amongst the environmental effects that affect DOAS measurements, multiple scattering should be included as the aim is detect smoke.*
*Page 5 line 9. Please explain what is instrumental light scattering, are authors referring to straylight here?*
*Equation (3). Please explain $A(\lambda,...)$ in more detail. What "..." is? In line 16 page 3 is defined as a ratio but it is not indicated of what magnitudes is this ratio.*
*Page 6 line 8. $SC_i$ is mentioned as the result of passive DOAS measurements, but this*

[Figure]

*is only true when I0((λ) is known, what is not usually the case for this kind of measure-ments. In fact this erroneous concept is propagated along the explanation affecting other parts of the work. This connect to figure 7, how can authors explain the exis-tence of negative optical thickness? (please notice as well that optical thickness is a non-dimensional magnitude).*

*Page 6 line 14, by spectral profiles authors mean spectral features? "quicker" and more important spectral signatures, is not a clear nomenclature, please rewrite this part.*

*Equation (6) magnitudes are not defined.*

*Equation (7) needs further explanation. Where the magnitude A(λ,...), Mie and Rayleigh scattering terms are?. What the new terms ∆SCi and ajλj are?*

*To eliminate I0(λ) is not as straightforward as authors mean, please notice that Iref is also affected by absorption and scattering and this is not mentioned in the step from equation (5) to (7), this is a part of what I meant in the previous comment of line 8 in this same page.*

*Page 7 line 6, Ring effect is not compensated but treat as an absorber by using a pseudo- absorption cross section. A small explanation, as this part of the work is mainly pedagogical, should be included about the calculation of this cross section and please include a reference.*

*Page 7 line 14. Please, if FFF does not apply directly DOAS technique itself, please indicate the differences with DOAS or in which way the technique is going to be used. This is an important part, especially when an important effort has been made to intro-duce DOAS technique.*

*Page 7 line 15. System is not measuring its surroundings but something in its sur-roundings, please specify what.'*

**Reply:**
The referee makes an extensive and thorough comment to our paper's Section 3. We had already agreed with to reformulate this section according to the first referee's com-ments. We will also take these into consideration in order to present a clearer picture of what we are trying to convey.

**Comment:**
*'Section "The Device" What is the Field of View of the telescope?'*

**Reply:**
It is 1,4° . We will include that value in the text.

**Comment:**
*'I understand that the percentage of false positives are low when compared to the high number of analysed spectra per day, but the reliability of this system should reside on the correct detection of forest fires, so the analysis of false positives is an important part that need to be improved. Authors should indicate if further work is going to be done in this line and indicate how this weakness of the system can be improved. It is interesting that most of false positives are due to presence of clouds, especially when DOAS can be also used to detect clouds and even aerosols. I don't know if authors are aware of the previous work in this area (Gielen et al., Atmos. Meas. Tech., 7, 3509–3527, 2014 www.atmos- meas-tech.net/7/3509/2014/ or , Wagner et al., Atmos. Meas. Tech., 7, 1289–1320, 2014 doi:10.5194/amt-7-1289-2014).'*

**Reply:**
This is a very important comment. Yes, the system's reliability is directly related to the number of false positives and yes, one of our research lines is the analysis of clouds and aerosols in order to classify false positives. This research is, however, conditioned by time and memory constraints, so it greatly depends on the success of the new spectral algorithms which are mentioned the text, and thus we chose not to include it in the text.

We greatly appreciate the referee's efforts in referencing papers that may help our work. For what it is worth, we had knowledge of the paper by Wagner et al., but not the other one.

We trust we have addressed the referee's doubts with the necessary clarity, but should the need arise, please do not hesitate to contact us.